# New Insights into Inflammatory Bowel Diseases from Proteomic and Lipidomic Studies

**DOI:** 10.3390/proteomes8030018

**Published:** 2020-08-10

**Authors:** Serena Longo, Marcello Chieppa, Luca G. Cossa, Chiara C. Spinelli, Marco Greco, Michele Maffia, Anna M. Giudetti

**Affiliations:** 1Department of Biological and Environmental Sciences and Technologies, University of Salento, via Monteroni 165, 73100 Lecce, Italy; serena.longo@unisalento.it (S.L.); lucagiulio.cossa@gmail.com (L.G.C.); chiaracarmela.spinelli@unisalento.it (C.C.S.); 2National Institute of Gastroenterology “S. de Bellis”, Institute of Research, Via Turi, 27, 70013 Castellana Grotte, Italy; marcello.chieppa@irccsdebellis.it; 3Department of Mathematics and Physics “Ennio De Giorgi”, University of Salento, via Monteroni, 73100 Lecce, Italy; marco.greco@unisalento.it

**Keywords:** Crohn’s disease, lipidomics, markers, proteomics, ulcerative colitis

## Abstract

Ulcerative colitis (UC) and Crohn’s disease (CD) represent the two main forms of chronic inflammatory bowel diseases (IBD). The exact IBD etiology is not yet revealed but CD and UC are likely induced by an excessive immune response against normal constituents of the intestinal microbial flora. IBD diagnosis is based on clinical symptoms often combined with invasive and costly procedures. Thus, the need for more non-invasive markers is urgent. Several routine laboratory investigations have been explored as indicators of intestinal inflammation in IBD, including blood testing for C-reactive protein, erythrocyte sedimentation rate, and specific antibodies, in addition to stool testing for calprotectin and lactoferrin. However, none has been universally adopted, some have been well-characterized, and others hold great promise. In recent years, the technological developments within the field of mass spectrometry (MS) and bioinformatics have greatly enhanced the ability to retrieve, characterize, and analyze large amounts of data. High-throughput research allowed enhancing the understanding of the biology of IBD permitting a more accurate biomarker discovery than ever before. In this review, we summarize currently used IBD serological and stool biomarkers and how proteomics and lipidomics are contributing to the identification of IBD biomarkers.

## 1. Introduction

Ulcerative colitis (UC) and Crohn’s disease (CD) are the two main clinically defined manifestations of the so-called inflammatory bowel diseases (IBD) characterized by chronic bouts of inflammation and remission in the gastrointestinal tract [1]. Despite some common clinical and pathological features including abdominal pain, diarrhea, and rectal bleeding, UC and CD can be distinguished by different genetic predisposition, risk factors, site, and nature of the inflamed lesions.

Indeed, while UC is characterized by a continuous inflammation, limited to the colon mucosa and submucosa, without involving the small bowel [2,3,4,5], CD is a transmural disorder characterized by a non-contiguous inflammatory pattern that may affect the entire gastrointestinal tract, from the oral cavity to the rectum (Figure 1) [6,7]. The CD’s primary clinical symptom is abdominal pain with weight loss; it can also present with extra-intestinal co-morbidities similar to those described in UC patients; however, CD patients may also present peri-anal complications (abscesses, fistulas, and strictures) [6].

Although the etiology of IBD is largely unknown, recent evidence suggests that in CD and UC, the gut-associated microbial ecosystem may play a crucial role for triggering and sustaining the chronic inflammatory response [8,9]. However, it is still unclear if the intestinal dysbiosis is the trigger or the result of a chronically inflamed intestine [10].

Under homeostatic conditions, a single layer of enterocytes represents an efficient physical, chemical and electrical barrier against luminal microbial community’s invasion but, when the integrity of the epithelium is compromised, luminal antigens, including pathobionts can translocate into the subepithelial compartment thus triggering and/or sustaining a dysregulated inflammatory immune responses [11,12]. The inflammatory trigger activates the synthesis of pro-inflammatory chemokines (chemoattractants) by epithelial cells with a robust influx of neutrophils into the tissue within hours of the damage [9]. Moreover, during the acute period of the inflammatory response, the mucosal milieu is enriched in inflammatory cytokines, metabolites of arachidonic acid and other pro-inflammatory mediators that activate the recruited leukocytes [9]. Once the inflammatory response has started, TNF-α, IL-1β, and other pro-inflammatory leukocyte products amplify the inflammatory response in a vicious circle can further damage the intestinal mucosa integrity (Figure 2).

Clinical presentation of IBD is non-specific and diagnosis is based on medical history, clinical, endoscopic, radiological, and histopathological findings. Although endoscopy remains the gold standard for the diagnosis and monitoring of IBD, the use of biomarkers in clinical practice is extremely important and supportive. IBD markers have been identified in colonic tissues, blood, stool, urine, and breath. Anyway, none of the markers nowadays available has enough high sensitivity and specificity to allow an early and differential diagnosis between UC, CD, and other colitis, limiting clinical efficacy. Therefore, clinical management of IBD would benefit from the identification of new, less invasive, targeted, and specific molecular biomarkers, thus improving early diagnosis and classification of these pathologies.

In the last decade, new technologies, such as genomics and proteomics, have been combined in the study of IBD and applied to the discovery of biomarkers of diagnosis, stratification, and treatment monitoring.

While genetic studies have been explored thoroughly in the context of IBD, proteomic and lipidomic research has greatly advanced in recent years offering the potential to identify previously unknown disease markers. This review aims to report the “state of the art” of biomarkers used in IBD and to report new analytical tools, based on lipidomics and proteomics, for the identification of new molecular markers.

## 2. Biomarkers in Inflammatory Bowel Disease

IBD is associated with an acute phase response, characterized by the production of several proteins, which may be detected in the serum or stools [13]. Some of them are related to coagulation and fibrinolysis (i.e., prothrombin, fibrinogen, plasminogen, factor VIII, complement system components), transport proteins (haptoglobin and ceruloplasmin), proteinase inhibitor (α1-antitrypsin and α1-antichymotrypsin), and cytokines [14]. Moreover, a variety of other proteins increased during acute phase inflammation, such as C-reactive protein (CRP), ferritin, fibronectin, and orosomucoid. Most of these markers, however, have not been largely studied in IBD or have produced conflicting results [13]. Cellular components of blood can be also involved in IBD and, in this respect platelet and leukocyte count and erythrocyte sedimentation rate (ESR) can be used as indirect measures of inflammation [13].

CRP is a human acute-phase protein produced during inflammation and the best laboratory marker in differentiating IBD from normal. However, the CRP response between UC and CD is rather heterogenic [15]. Thus, while CD is associated with a strong CRP response, UC has only a modest to absent correlation with this marker [16].

ESR has been reported to be a marker to differentiate between IBD and normal, with a higher value in CD than UC [13].

A serological characteristic of IBD patients is the presence of several particular antibodies [17,18]. Serum markers, against microbial antigens, include perinuclear anti-neutrophil cytoplasmic antibody (pANCA), anti-*Saccharomyces cerevisiae* antibody (ASCA), anti porin (anti-OmpC) a protein of the outer membrane of *Escherichia coli*, anti-Cbir1 Flagellin (anti-CBir1) and anti-*Pseudomonas fluorescens*-associated sequence I-2 (anti-I2).

Although some of these antibodies allow differentiating UC from CD, as in the case of pANCA [19,20], or having a high specificity in identifying CD, such as ASCA [21], their poor sensitivity limits their use [22,23]. Nevertheless, the coupled use of the two markers increases their diagnostic potential [20,24,25].

Anti-OmpC, anti-CBir1, and anti-I2 have been reported to be more specific for CD than UC suggesting that their use can help differentiation between IBD subtypes [26,27,28,29,30,31].

Anti-pancreatic antibodies have been detected in few UC patients (2–6%) and, antibodies against p53 protein, the product of a gene which has been frequently mutated in human cancers, have been also measured in the serum from a low percentage of UC patients (about 9.3%) [32].

The search for stool biomarkers is also an explored field in IBD. Fecal calprotectin (FC) represents a protein of neutrophil granulocytes with antimicrobial activity. FC is not specific for IBD but can be used as an indicator of disease exacerbation [33]. Moreover, FC increases in neoplasia, infections, and polyps [34], as well as during non-steroidal anti-inflammatory treatments [35], and with aging [36]. Lactoferrin is expressed by neutrophil during inflammation. Although lactoferrin can distinguish IBD from other forms of colitis, it does not distinguish CD from UC [37]. Table 1 summarizes the biomarkers commonly used for IBD.

## 3. New IBD Markers from Mass Spectrometry

Over the past decade, mass spectrometry (MS) has emerged in the biomedical setting as a valuable tool for the diagnosis of several physio-pathological conditions leading to a better understanding of the molecular mechanisms and functional networks behind human diseases [38,39,40,41]. Among different molecules that can be detectable with MS, metabolites, proteins and lipids have been provide useful information in clinical. This has generated a large amount of data that led to the creation of large international projects (https://www.hupo.org/B/D-HPP), and large repository (https://www.ebi.ac.uk/pride/). The approach is now evolving to embrace novel in vivo clinical applications for the rapid and real-time analysis of human samples [42,43]. Proteomic and, to a lesser extent, lipidomic studies have already been successfully used to investigate IBD pathomechanisms, including the inflammatory response, epithelial barrier function, and gut microbiome. These methodological approaches could effectively accelerate the development of novel complementary biomarkers and could help the monitoring of the treatment response to facilitate a personalized medicine approach to IBD. Below, we report the main researches carried out on protein and lipid analysis in IBD by MS approaches.

### Proteomic Analyses

Proteomics is an analytical technique used for the accurate identification of unknown analytes in different biological samples including, serum/blood, tissue samples and feces. Currently, proteomics relies mainly on top-down and bottom-up approaches for protein analysis and further divided into discovery proteomics and targeted proteomics. A large selection of methods and instruments can now be used to explore the modular and spatial organization of the proteome at a multidimensional level [44,45].

Several proteomic studies have successfully been aimed to study the etiology of IBD and have attempted to establish the correct diagnosis of CU and CD patients (Table 2).

The great majority of these studies have successfully employed two-dimensional electrophoresis (2-DE), and matrix-assisted laser desorption/ionization-time of flight (MALDI-TOF-MS) for protein analysis and identification in IBD. Although the results of these studies revealed a complex network of molecules and processes modulated in IBD samples, from a technological point of view these approaches suffer for a lower sensitivity compared to more recent studies conducted by using liquid chromatography (LC)-MS/MS approaches.

The first study based on MALDI-TOF-MS was conducted by Barceló-Batllori et al., about 20 years ago [46]. The study aimed to identify proteins involved in IBD using colonic epithelial cells from CD and UC patients The authors reported an overabundance of indoleamine-2,3-dioxygenase enzyme in both CD and UC samples as compared to normal mucosa, thus highlighting involvement of tryptophan metabolism in IBD [46]. Being tryptophan a precursor of serotonin, an altered metabolism of this amino acid in intestinal enteroendocrine cells can be associated with alteration in intestinal secretion and mobility.

In another study, the proteomic profile of colon mucosa from UC patients, in the acute phase, was examined by MALDI-TOF-MS to search for possible markers associated with disease exacerbation. About 222 signals were found differentially expressed in UC with respect to non-inflamed tissue. Among changed features, 43 individual proteins were identified and linked to energy metabolism and oxidative stress [47].

An attempt to distinguish between CD and UC has been made by histology-directed MALDI profiling [48,49]. The advantage of both studies was to perform a diagnosis based only on *m/z* (mass to charge ratio) signatures directly on formalin-fixed, paraffin-embedded tissues from hospital pathology. One of these studies [48] reported significant discriminatory peaks in both inflamed and uninflamed colonic submucosa from UC and CD. The methodology revealed 8 peaks of interest and among these, 5 were individually considered as “good classifiers”. The other study [49], comparing histological layers from UC and CD patients, identified differences in the proteomic profile between UC and CD thus improving the accuracy of diagnostic and the management of IBD.

There is no doubt of the usefulness of this in situ approach that can now more easily be combined with instruments that have increased in efficiency and resolution.

A fine histological evaluation of colonic tissue specimens from UC and CD patients was performed using laser microdissection and LC-MS/MS [50]. In this study, a higher abundance of proteins related to neutrophil activity and damage-associated molecular patterns was measured in CD with respect to UC patients. Interestingly, the authors identified a protein group (Aldo-keto reductase family 1-member C3) that was present in almost all CD and absent in UC samples, thus indicating an involvement of the steroidogenic pathway in the etiopathology of CD.

By using 2-D electrophoresis (2-DE) followed by MALDI-TOF-MS, the involvement of mitochondrial dysfunction in the pathogenesis of IBD was also demonstrated. Among UC active, UC inactive, nonspecific colitis, and normal colon mucosa, authors reported a down-regulation of mitochondrial heat-shock protein 90, heat-shock protein 60, H1-transporting two-sector ATPase, prohibitin, malate dehydrogenase, voltage-dependent anion-selective channel protein 1, thioredoxin peroxidase and thiol-specific antioxidant [51].

Focusing on immune-cell characteristics, a study conducted on peripheral blood mononuclear cells allowed to discriminate UC from CD patients. Sample proteins were separated by 2-DE and subjected to in-gel tryptic digestion followed by MALDI-TOF-MS protein identification. The study underlined a different level, between UC and CD, of 7 proteins associated with inflammation oxidation/reduction, the cytoskeleton, endocytic trafficking and transcription [52]. With the same experimental approach, Shkoda and coll. [53] reported changes in 9 proteins between CD and UC intestinal epithelial cells, including Rho-GDP dissociation inhibitor alpha, a key regulator of cell signalling, that was up-regulated in CD and UC patients. Additionally, intestinal epithelial cells from inflamed compared to noninflamed tissue regions of UC patients showed a significant change in the abundance of programmed cell death proteins and annexin 2A, this last protein being involved in the regulation of cell growth and signal transduction pathways.

A very recent study, conducted by MALDI-TOF-MS, analyzed the protein composition of stools from IBD patients and controls. The study highlighted differently expressed proteins between controls and IBD patients, with IBD-associated over-expressed proteins such as immunoglobulins and neutrophil proteins, and under-expressed proteins comprising proteins of the nucleic acid assembly or those related to cancer risk [54].

Also, two recent works, analyzing the proteome from interstitial samples, allowed discrimination among UC, CD and controls. One of these studies [55], using LC-MS/MS, provided novel insights into the molecular pathogenesis of IBD by reporting the involvement, in IBD, of cell adhesion proteins such as CD38, whose abundance was higher in both CD and UC patients than in controls, and of proteins regulating blood pressure, such as angiotensin-converting enzymes 1 and 2 that showed higher levels in CD than in UC [55].

The other study [56], conducted also by LC-MS/MS, analyzed mucus samples from colon biopsies from UC patients with ongoing inflammation or in remission. The study highlighted a reduced number of sentinel goblet cells and attenuation of the goblet cell secretory response to a microbial challenge in the active UC with respect to healthy patients. Moreover, the study also demonstrated that the alteration in mucus samples included a reduction of the SLC26A3 apical membrane anion exchanger, which supplies bicarbonate required for colonic mucin barrier formation [56].

In previous years, the molecular signature of intestinal epithelial cells from UC or CD colonic specimens and non-inflammatory controls was investigated by gel-based stable-isotope label technologies (2D-DIGE and ICPL LC-MS/MS) followed by immunoblot validation [57]. This methodology, based on the incorporation of stable isotopes into proteins, allows a quantitative profiling within complex biological mixtures. Authors described changes in several molecules involved in the extracellular matrix, metabolic rewiring and autophagy that characterize quiescent UC epithelial cells thus helping the understanding of the complex mechanisms associated with IBD. In particular, authors reported, in UC patients with respect to non-inflamed controls, an increment of a specific enzyme involved in the degradation of branched-chain fatty acids and of the level of p62 and myosin heavy chain both proteins involved in macro-autophagy. In the same year, a study based on LC-MS/MS analyzed the tryptic digestion of proteins from mucosal biopsies of IBD children. In the study, a panel of 5 different proteins was established to discriminate IBD from control patients and a 12-protein panel was proposed to distinguish CD from UC patients [58]. The authors found decreased FABP5 protein levels in children with IBD compared with control patients, and energy metabolism (inorganic pyrophosphatase, visfatin and UDP-glucose 6-dehydrogenase) was altered in IBD. Proteins involved in fatty acid metabolism (leukotriene A-4 hydrolase, tricarboxylate transport protein, trifunctional enzyme and delta (3,5)-delta (2,4)-dienoyl-CoA isomerase) were up-regulated in CD compared to UC pediatric patients.

A few years earlier, a proteomic study by LC-MS/MS revealed some aspects of the molecular mechanism of IBD response to Infliximab, an anti-TNF agent commonly used to control inflammation in IBD patients. The study, conducted on blood samples and biopsies obtained from responders and non-responders UC patients treated in vitro with or without Infliximab demonstrated that the drug response is associated with reduced monocyte activation and lesser chemokine secretion [59]. In the same year and with the same experimental approach [60], a total of 46 proteins were found significantly changed between controls and UC mucosal biopsies. In particular, lactotransferrin was 219 times more abundant in the UC group. The relative abundance of lactotransferrin was also correlated to the severity of tissue inflammation in patients with UC, as determined by the colon inflammation grade score based on histology [60].

The subcellular distribution of different proteins can help the understanding of the grade of cell compromising during pathology. To this respect, by LC-MS/MS, up-regulated proteins in CD patients, with respect to healthy donors, were found in the nuclear fraction (histones and ubiquitin), in the cytosolic fraction (Tryptase alpha-1 precursor) and in the membrane fraction (ATP synthase subunit beta and heat shock 70k Da protein 5) thus indicating that IBD alterations can be detected at different levels, with probable alterations in gene expression.

## 4. Lipidomic Analyses

Lipidomics is a science that allows the systems-level analysis of lipids and their interactions [62,63]. Lipidomics has significantly improved diagnostic medicine as well as treatment options for many diseases [63,64,65,66]. Nowadays, the lipidomic analysis relied on the use of atmospheric pressure ionization MS either without separation or coupled with gas-phase or liquid-phase separation techniques such as gas chromatography-TOF-MS (GC-TOF-MS), (ultra)high-performance liquid chromatography ((U)HPLC) or (ultrahigh-performance) supercritical fluid chromatography ((UHP)SFC). Electrospray ionization (ESI) coupled with MS (ESI-MS) is, by the far, the most frequently used analytical technique for map a lipid landscape due to easy coupling with liquid-phase separation techniques and applicability for a wide range of lipids.

Lipidomics has been applied in several IBD biomarker identification studies (Table 3).

By using a lipidomic approach by LC-MS/MS, a lipidomic study was conducted on colon and blood tissue samples from UC patients and healthy controls. The study reported a reduced de novo synthesis of sphingolipids and an increased amount of lactosyl-ceramides in inflamed colon tissues, with respect to controls. Furthermore, in human plasma from UC patients, in comparison with healthy subjects, significant changes were measured in the level of several sphingolipids, free fatty acids, lyso-phosphatidylcholines (lyso-PC) and triglycerides being these changes dependent from the disease severity. Thus, the study pointed out an important role for de novo synthesis of sphingolipids in UC trig [67].

A study performed by combining GC-TOF-MS and UHPLC-MS on colon biopsies reported an altered balance between pro- and anti-inflammatory lipid mediators between treatment-naïve UC patients, deep remission UC patients, and healthy controls. In particular, increased levels of ω-6-related oxylipins and decreased levels of ω-3-related endocannabinoids were correlated with UC debut [68].

In a case-control study, lipid mediators were determined by LC-MS/MS in mucosal biopsies taken from UC relapsing patients [69]. Inflamed mucosa showed increased levels of seven eicosanoids (prostaglandin (PG) E2, PGD2, thromboxane B2, 5-hydroxyeicosatetraenoic acid (HETE), 11-HETE, 12-HETE, and 15-HETE) that correlated with the degree of inflammation.

Nano-ESI MS/MS was used to compare the lipid composition of intestinal mucus from patients with UC, CD and control subjects. Patients with UC showed significantly less phosphatidylcholine (PC) and lyso-PC in rectal mucus compared to subjects from the other groups and attributed the increased susceptibility to luminal contents of UC patients to the low amount of protective mucus PC [70]. A study conducted with the same methodology (nano-ESI-MS/MS), reported alteration in the phospholipid concentration of intestinal mucus of patients with UC in comparison with CD and healthy controls. In particular, the concentration of PC, lyso-PC, and sphingomyelin was significantly lower in mucus specimens from UC than CD patients and healthy controls. Moreover, in UC patients, fatty acids of PC showed a higher grade of saturation than CD patients and healthy controls and UC patients showed also a higher LPC-to-PC ratio [71].

By using LC-ESI-tandem-MS, a comparison of plasma lipid profiles from UC, CD, and healthy controls was made [72]. This study reports a total of 33 individual lipid species, mainly belonging to the ether lipids and plasmalogens, which were significantly and negatively associated with CD and only five lipid species significantly correlated with UC.

Analysis of stool samples from IBD and controls subjects carried out by GC-MS and LC-QTOF-MS, demonstrated increased levels of diacylglycerol and *n*-acyl-phosphatidylethanolamines in IBD patients when compared with healthy individuals, while urobilin, PC, urobilinogen, phosphatidic acid phosphatidylserine, and ceramide were decreased [73].

By UPLC-MS/MS, Scoville and colleagues [74] showed that the serum metabolomic profile in IBD patients reflected differences in several lipids, amino acids, and tricarboxylic acid cycle-related metabolites when compared to the healthy controls [74]. Moreover, another study conducted on plasma samples using LC-TOF-MS reported differences in the lipid profile from IBD and healthy individuals. PC, lyso-PC, and fatty acids were significantly changed among pathological samples suggesting changes in phospholipase A2 and arachidonic acid metabolic pathways. Variations in the level of cholesteryl esters and glycerophospholipids were also reported [75].

## 5. Conclusions

The diagnosis of IBD currently depends on clinical, endoscopic and histological evaluations.

Several diagnostic sera and fecal inflammatory biomarkers are currently used in IBD clinical practice but often they require coupling with more specific and invasive diagnostic approaches.

MS-based methodologies represent a powerful tool for identifying and quantifying several thousand metabolites with different physicochemical properties. This has a number of advantages including the possibility to define functional networks between different analytes. These approaches have been applied with success for the analysis of protein and lipid component of IBD samples.

Importantly, MS-studies conducted by histology-directed MALDI-TOF-MS [48,49] has been used to map the localization of peptides directly on-tissue. More recently, the application of LCM in combination with LC-MS/MS overcame the limitations of pioneering imaging-approaches combining the high-throughput capacity of LC-MS/MS with the analysis of distinct tissue regions.

Here emerges that IBD, besides involving proteins of immunity and of inflammatory responses [50,54,55,60,67], can involve unexpected proteins such as those involved in the regulation of blood pressure [55] and cancer [54].

The pro-inflammatory phenotype of IBD, analyzed by a lipidomic point of view, defines a change in the omega-6/omega-3 ratio [68,69] with the appearance of harmful lipid species, such as plasmalogens, lysophospholipids, and ethers lipids [70,71,72,74,75]. These studies can put the basis for targeted therapies, as reported for ω-3 fatty acid supplementation that has been demonstrated to induce beneficial effects, in UC intestinal epithelial, on tight junction permeability and oxidative stress [76,77].

Moreover, the dysregulation of metabolic enzymes involved in lipid synthesis, with important repercussions in cell turnover following epithelial barrier disruption, is an interesting aspect emerging from lipidomic studies.

These aspects, together with the findings that IBD induces derangement in mucus component, cytoskeleton as well as intestinal cell functions [52,56,57] can help the understanding of the IBD pathology and strategical interventional approaches.

MS studies have allowed identifying new metabolic pathways deregulated in IBD, such as autophagy, glycolysis, tricarboxylic acid cycle, mitochondrial, and energy metabolism [47,51,52], and could furnish the basis for set-upping new IBD treatments.

A relationship between proteomic and lipidomic results can likely be expected. In fact, changes in the expression and/or activity of enzymatic proteins involved in lipid synthesis can account for specific changes in lipidomic profiles. In this perspective, a different expression of proteins involved in fatty acid uptake and lipid synthesis in ileum and colon of IBD patients as been reported [78]. On the other hand, lipids play a distinct role as a second messenger in signaling pathways and are essential for physiological properties of cell membranes. Lipid metabolism and signaling are suggested to play important roles in inflammation with significant implications for IBD [70,71,72,74,75]. Thus, qualitative and quantitative alterations in their structure may alter specific signal pathways that, in turn, generate changes in proteomic signals.

The information deriving from the proteomic and lipidomic analysis could be used to make accurate diagnostic and prognostic tools to differentiate patient groups and predict response to treatment. In addition to helping the correct diagnosis and identification of an adequate therapeutic strategy, the study of the entire proteome and lipidome of tissue could allow identification of the course of the disease and the identification of new therapeutic agents.

Taken together, MS-based analyses are effective methodologies for a rapid mapping of protein and lipid distribution in tissues, with little or no sample preparation. In conclusion, the identification and quantification of proteins and lipids species throughout MS approaches could explain the etiology of the disease, leading to improved treatment strategies. In the future, correlating serologic markers with proteomics and lipidomic phenotypes will enhance our understanding of the pathophysiology of IBD. Thus, the association of lipidomic and proteomic analyses allows raising the level of investigation of IBD.

## Figures and Tables

**Figure 1 proteomes-08-00018-f001:**
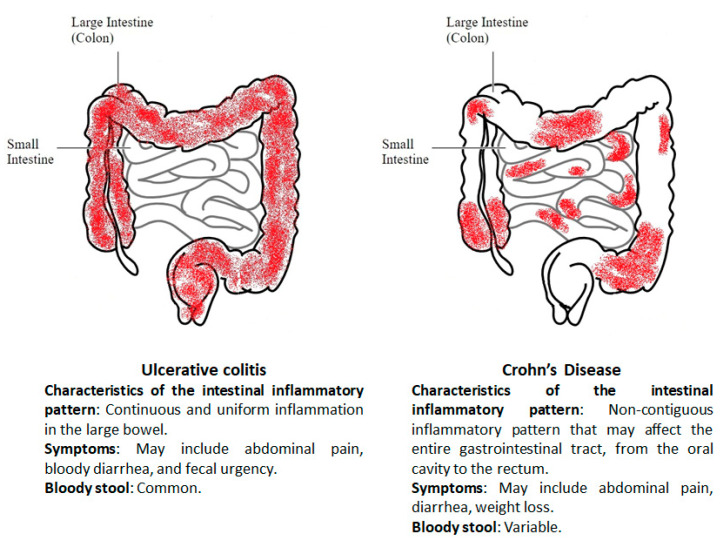
Different localization and main symptoms in ulcerative colitis and Crohn’s disease.

**Figure 2 proteomes-08-00018-f002:**
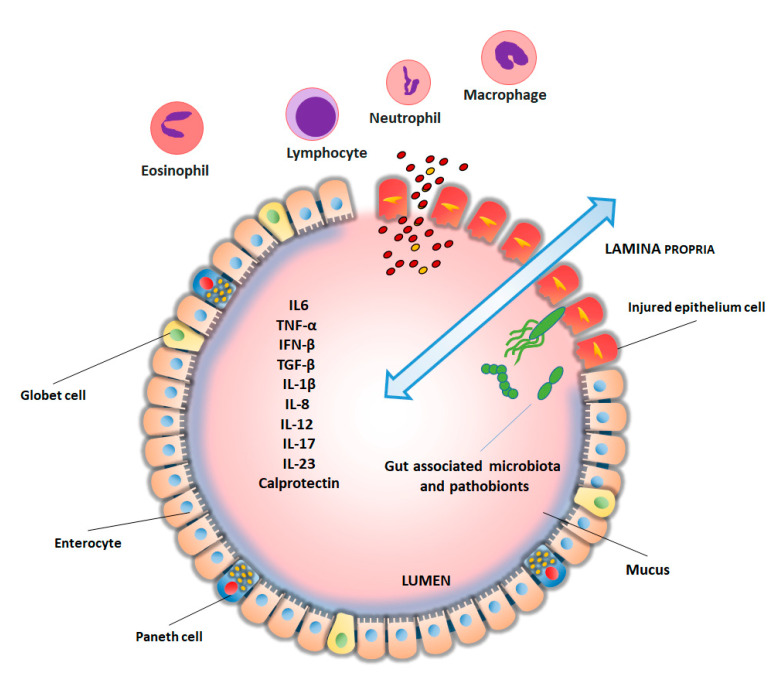
Inflammatory mechanisms and biomarkers in inflammatory bowel disease (IBD). The intestinal mucosa is mainly composed of enterocytes, goblet cells, and Paneth cells. A layer of mucus overlies the epithelium and limits contact between bacteria and cells. Changes in the gut microbiota and the disruption of epithelial barrier function can trigger a dysregulated mucosal immune response and promote the synthesis of inflammatory mediators (TNF-α, IFN-β, TGF-β, IL-1β, IL-6, IL-8, IL-12, IL-17, and IL-23) with the recruitment of granulocytes (neutrophils, eosinophils), lymphocytes and macrophages. Moreover, disruption of epithelial barrier can initiate bilateral passage of cellular components and inflammatory mediators (double arrow). Inflammatory mediators can be measured directly in colonic biopsies or upon release into the gut (i.e., fecal calprotectin), but acute phase response is detectable in the serum and blood.

**Table 1 proteomes-08-00018-t001:** Biomarkers in inflammatory bowel disease.

Biomarker	Source	Ref.
C-reactive protein	Serum	[16]
Erythrocyte sedimentation rate	Blood	[13]
Anti-*Saccharomyces cerevisiae* (ASCA)	Serum	[21]
Perinuclear antineutrophil cytoplasmic antibody (pANCA)	Serum	[19,20]
Anti-porin (Anti-OmpC)	Serum	[26,27,28]
Anti-Cbir1 Flagellin (anti-CBir1)	Serum	[29]
Anti-*Pseudomonas fluorescens*-associated sequence I-2 (Anti-I2)	Serum	[30]
Fecal calprotectin	Stool	[32]
Lactoferrin	Stool	[36]

**Table 2 proteomes-08-00018-t002:** Proteomic applications in inflammatory bowel disease.

Key Findings	Biological Sample	Separation	Instrument	Ref.
Higher abundance, in IBD vs. normal mucosa, of indoleamine-2,3-dioxygenase.	Colonic epithelial cells from ulcerative colitis (UC) and Crohn’s disease (CD) patients	2-DE	MALDI-TOF-MS	[46]
Distinct profile in UC vs. controls of proteins involved in energy metabolism and oxidative stress.	Colonic biopsies from UC and controls	2-DE	MALDI-TOF-MS	[47]
Identification of a different proteomic signature between CD and CD.	Colonic mucosal and submucosal layers from CD and UC	On-tissue analysis	Histology-directed MALDI-TOF-MS	[48]
Identification of a different proteomic signature between CD and CD.	Histologic layers from UC and CD	On-tissue analysis	Histology-directed MALDI-TOF-MS	[49]
Higher abundance in CD compared to UC of the Aldo-keto reductase family 1 member C3 (AKR1C3) protein.	Colonic tissue specimens from UC and CD	Laser microdissection and chromatography	LC-MS/MS	[50]
Down-regulation of different mitochondrial proteins involved in energy generation and stress response.	Colonic biopsies from UC, nonspecific colitis patients and controls	2-DE	MALDI-TOF-MS	[51]
Identification of a different UC and CD protein signature, comprising proteins associated with inflammation, oxidation/reduction, the cytoskeleton, endocytic trafficking and transcription.	Peripheral blood mononuclear cells from UC and CD	2-DE	MALDI-TOF-MS	[52]
Up-regulation of Rho-GDP dissociation inhibitor alpha in CD and UC patients. In UC vs. noninflamed higher level of programmed cell death proteins and annexin 2A.	Human primary intestinal epithelial cells obtained from CD, UC and control patients	2-DE	MALDI-TOF-MS	[53]
Higher level in IBD of immunoglobulins and neutrophil proteins. Lower level in IBD of the nucleic acid assembly proteins or OLFM4, ENPP7, related to cancer risk.	Stools from IBD and control patients	Peptide analysis and chromatography	MALDI-TOF-MS/MS and LC-MS/MS	[54]
Higher level in CD and UC of CD38 and angiotensin-converting enzymes 1 and 2.	Colonic biopsies from UC, CD and controls	Chromatography	LC-MS/MS	[55]
Reduced amount in active UC of mucin MUC2 and SLC26A3.	Mucus samples from inflamed or in remission UC patients	Chromatography	LC-MS/MS	[56]
Changes in UC and CD of proteins of extracellular matrix, cytoskeletal, cellular metabolism, and autophagy.	Colonic biopsies from UC patients and controls	2DE and chromatography	MALDI-TOF-MS and ICPL-LC-MS/MS	[57]
5 different proteins discriminate IBD from control patients and a 12-protein panel was proposed to distinguish CD from UC patients.	Mucosal biopsies from IBD children and controls	Chromatography	LC-MS/MS	[58]
Increased amount in UC of proteins involved in the innate immune system.	Colonic biopsies from controls and UC patients	Chromatography	LC-MS/MS	[60]
Up-regulation in CD patients of nuclear histones and ubiquitin, of cytosolic tryptase alpha-1 precursor and of membrane ATP synthase subunit beta and Heat shock 70kDa protein 5.	Subcellular fractions of intestinal epithelium cells from healthy donors and CD patients	Chromatography	LC-MS/MS	[61]
**Abbreviations:** two-dimensional electrophoresis (2-DE), matrix assisted laser desorption ionization-mass spectrometry (MALDI-MS), time of flight (TOF), difference gel electrophoresis (DIGE), liquid chromatography-mass spectrometry/mass spectrometry (LC-MS/MS), Isotope-Coded Protein Labeling (ICPL).				

**Table 3 proteomes-08-00018-t003:** Lipidomic applications in inflammatory bowel disease.

Key Findings	Biological Samples	Instrument	Ref.
In inflamed colon tissue *de novo*-synthesis of sphingolipids was reduced, whereas lactosylceramides were increased. Plasma sphingolipids, free fatty acids, lyso-PC and triacylglycerols changed significantly in UC in comparison to healthy controls.	Blood and colon tissue samples from UC patients and healthy controls	LC-MS/MS and LC-QTOF-MS	[67]
Debut of UC is associated with increased levels of ω-6-related oxylipins and decreased levels of ω-3-related endocannabinoids.	Colon biopsies from treatment-naïve UC patients, deep remission UC patients, and healthy controls	GC-TOF-MS and UHPLC-M	[68]
Levels of PGE2, PGD2, TXB2, 5-HETE, 11-HETE, 12-HETE and 15-HETE were significantly elevated in inflamed mucosa and correlated with severity of inflammation.	Mucosal biopsies from relapsing UC	LC-MS/MS	[69]
Significantly less PC and lyso-PC in patients with inactive UC compared to CD and control subjects.	Rectal mucus from patients with UC, CD and healthy controls	Nano-ESI-MS/MS	[70]
Lower PC concentration in UC compared to CD patients and controls. Independent of disease activity, patients suffering from UC showed an increased saturation grade of PC fatty acid residues and a higher lyso-PC-to-PC ratio.	Colonic mucus from UC, CD and healthy controls	Nano ESI-MS/MS	[71]
Lipid species belonging to ethers and plasmalogens were significantly changed in CD patients compared with controls. Only 5 lipid species significantly differed between UC and controls.	Plasma from IBD and controls	LC-ESI-MS/MS	[72]
Biogenic amines, amino acids, lipids, were significantly increased in IBD, while others, such as two B group vitamins, were decreased in IBD compared to healthy subjects.	Stools from UC, CD and healthy controls	GC-MS and LC-QTOF-MS	[73]
Several lipid-, amino acid-, and tricarboxylic acid cycle-related metabolites were significantly altered in CD. Instead, only 5 metabolites decreased in UC with respect to control subjects.	Serum from UC, CD and healthy controls	UPLC-MS/MS	[74]
PC, lyso-PC and fatty acids were significantly changed among pathological samples. Variations in the levels of cholesteryl esters and glycerophospholipids were also found.	Plasma from IBD and healthy controls	LC-TOF-MS	[75]
**Abbreviations:** phosphatidylcholine (PC), lyso-phosphatidylcholines (lyso-PC), prostaglandin (PG), tromboxane 2 (TXB2), hydroxyeicosatetraenoic acid (HETE).			

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
