# Peer review of "New Insights into Inflammatory Bowel Diseases from Proteomic and Lipidomic Studies"

_proteomes, 2020, doi:10.3390/proteomes8030018_

Round 1

Reviewer 1 Report

Longo et al presents a review on IBD markers and potential proteomics and lipidomics approaches. The topics is relevant and timely but the presentation and review-style needs improvements to be acceptable for publication. While the manuscript text is clear, it contains several minor English errors that should be corrected with a proper proof and style check. As a review, the biggest issue is that most of the sections read much like a list, lots of descriptive paragraphs with little added value, interpretation or connection between the different studies. It is also not helpful that there is only a single figure for this paper, while for a review as long as this one at least three figures are expected. A significant part of the manuscript is not novel, the first part is a general but long summary on IBD. While the topic is justified given the readership of Proteomics, probably less details on IBD would help. Especially as there are many other reviews on the topics of epithelial structure, antibodies, blood and serum markers. For the main focus of the paper, ie, the proteomics and lipidomics sections, it should be clear what is the novelty from other reviews (eg. https://www.ncbi.nlm.nih.gov/pmc/articles/PMC6163330/). The discussion section is good but could be expanded a lot. Have findings been replicated in multiple studies? Is there a better or worse methodology? Have any of the findings led to treatments or patient stratification methods? Pros and cons of different approaches? None of this is covered. The other main issue is that almost there is no discussion of combined multi-omics approaches, despite some multi omics paper being referenced. Also, currently the title containing multi-omics is not justified. The paper is focusing on only proteomics and lipidomics, which is fine but multi-omics often means mass-spec analysis combined with for example sequencing approaches. This review is limited from that but it is not an issue, a focused review on how mass-spec approaches could contribute to IBD marker development is very much needed, and would be good to have after a careful revision of this manuscript, as it already contains most of the major papers of the field.

A formatting or technical major issue is that the references/citation in the proteomics section don’t match what they are supposed to be referencing.

Other comments:

  • Figure 1 is difficult to read due to small text and poor quality
  • On line 54/55 the reference to UC being a Th2 response is somewhat oversimplified and out of date. It could be noted that this is an area of some contention. The same goes for lines 60/61 regarding CD - for example there are certainly references stating that Il17 is important in UC as well as CD. Also, find some newer references here.
  • Reference 6 has no title in the bibliography, making it difficult to identify.
  • Line 87 - These are not the only cells of the intestinal epithelium. Also it should perhaps be noted that Paneth cells are only found in the small intestine.
  • The section titled 'Epithelial biology in the gastrointestinal system' starting on line 81 is too long and references mostly very old papers.
  • Table 2 line 270 - This table would benefit from reformatting into an easier to digest layout. For example it would be nice to see easily which proteins are increased/deceased in UC/CD.
    • Within this table the terms 'overexpression' and 'expression' are used a few times to refer to protein levels. This is incorrect and misleading.
    •  

Author Response

Other comments:

  • Figure 1 is difficult to read due to small text and poor quality
    • Accordingly, we changed Figure 1 splitting it into two figures that can better highlight their content.

  • On line 54/55 the reference to UC being a Th2 response is somewhat oversimplified and out of date. It could be noted that this is an area of some contention. The same goes for lines 60/61 regarding CD - for example there are certainly references stating that Il17 is important in UC as well as CD. Also, find some newer references here.

  • To make the manuscript more fluid, and to accomplish the requests of reviewers, some parts of the manuscript have been eliminated, therefore, the part to which the reviewer refers is no longer present in the revised version of the manuscript.

  • Reference 6 has no title in the bibliography, making it difficult to identify.
    • we have now checked all bibliographic voice.

  • Line 87 - These are not the only cells of the intestinal epithelium. Also it should perhaps be noted that Paneth cells are only found in the small intestine.
  • The section titled 'Epithelial biology in the gastrointestinal system' starting on line 81 is too long and references mostly very old papers.
    • We simplified the text of the manuscript by deleting the entire paragraph 'Epithelial biology in the gastrointestinal system' that does not furnish substantial information in the context of the review.

  • Table 2 line 270 - This table would benefit from reformatting into an easier to digest layout. For example it would be nice to see easily which proteins are increased/deceased in UC/CD.
    • According to the reviewer’s suggestion, we have better organize Table 2, not only by adding markers increased/decreased in UC vs CD but also by better describing the technology used in the study.

  • Within this table the terms 'overexpression' and 'expression' are used a few times to refer to protein levels. This is incorrect and misleading.
    • We thank the reviewer for the note. Accordingly, we made corrections in the text.

Reviewer 2 Report

GENERAL

This review attempts to offer an overview of inflammatory bowel disease (IBD) pathophysiology and the associated immune/inflammatory biomarkers that have been identified in the past and are still used in clinical practice, as well as the potential new biomarkers derived from proteomic and lipidomic analyses based on technologically advanced omic approaches. The content of the review is not particularly well balanced, and it does not properly translate the title, “Multi-omic analysis and new perspectives in inflammatory bowel diseases”. In fact, the first half of the manuscript unevenly and superficially describes a variety of biological abnormalities described in ulcerative colitis (UC) and/or Crohn’s disease (CD) patients, while the second half describes proteomic and lipidomic changes more recently reported in the literature. What new perspectives are offered is not well outlined. The two parts of the manuscript unambiguously reflect the different areas of expertise of the authors, which are knowledgeable in proteomic and lipidomic analyses, but not in pathophysiological and clinical aspects of IBD. A series of comments are offered to the authors for their consideration.

SPECIFIC

A central issue that confounds the reader is the way that the authors use the words “biomarker” and “marker” in this review. In principle, any measurement of any biological product found in IBD could be defined as a potential “marker” of the disease if it can help in the differential diagnosis or clinical activity of UC or CD. In practice, however, the word “biomarker” is traditionally restricted to those products that have been carefully tested, measured and demonstrated to be actually useful in the clinical setting. Thus, most of the “markers” listed in Table 1 are not true biomarkers, such as ferritin, fibronectin, haemopexin, … sialic acid,.. complement,…interleukin 2,…TL1alpha,…platelet count,…etc. etc., and only some are, such as CRP, … ASCA, ANCA,…calprotectin. Lumping them all together as “biomarkers” is improper and misleading.

Page 2:

The sentence “From an immunological standpoint, …. by natural killer T cells” is no longer valid or correct. In fact, the importance of IL-13 to UC pathogenesis has been dismissed by clinical trials showing no clinical benefits by the administration of anti-IL-13 monoclonal antibodies.

The paragraph on the enteric nervous system is disproportional long compared to its known relevance to IBD pathophysiology.

Page 3 and 4:

The whole section “2. Epithelial biology in the gastrointestinal system” is poorly structured and provides scattered and incomplete information. It tries to provide an all-inclusive picture of what goes on in IBD pathogenesis, which should not be part of the scope of the review.

The whole sentence “Moreover, epithelial cells… amplify the inflammatory response” is misinforming because it suggests that epithelial cells are the main source of TNF-alpha, which is not the case (macrophages, monocytes and T cells are), and that other pro-inflammatory products are subsequently released; in reality, there are no studies proving which cytokines and cells comes first or which follow, and it is likely that most pro-inflammatory molecules are released concomitantly.

Page 4 to 8:

The sections “3. Antibodies, blood and serum biomarkers” and “4. Stool markers” suffer from the same drawbacks and limitations of the previous section 2. It is also poorly structured and provides scattered and skewed information. Moreover, some of the information is no longer of any value such as the ALCA, ACCA, ASMA and AMCA, because of lack of clinical value and clinical use.

Page 8 to 13:

The section “5. New biomarkers from mass spectrometry” represents a list of more recent findings in proteomic and lipidomic analyses, the areas of expertise of the authors. Although detailed and more coherent, does not really provide new insights on how these studies can improve current knowledge of IBD pathophysiology.

ADDITIONAL COMMENTS

Figure 1:

The legend states “CD… is usually located in the lower part of the small bowel and the upper colon”, but this is not shown in the figure, where the terminal ileum and ascending colon are not inflamed.

DAMPS are mentioned in the legend but nowhere in the text.

“injured epithelium cell” should be “injured epithelial cell”.

“Goblet cells” should be “goblet cells”.

Page 2:

“UC is characterized by a continuous inflammation often limited to the epithelial mucosa”: there is no such a thing as the “epithelial mucosa”, but eventually “mucosal epithelium”; the sentence should actually read “UC is characterized by a continuous inflammation limited to the mucosa and submucosa”.

Table 1:

Each listed marker should be referenced as in Table 2.

Line 144:

“accomplished” should be “accompanied” or “associated with”.

References:

Many of the references, particularly those that cite findings on cytokines and other immune/inflammatory products in IBD, are from the 1990’s, and many better and more recent comprehensive publications on these topics are available.

Author Response

SPECIFIC

A central issue that confounds the reader is the way that the authors use the words “biomarker” and “marker” in this review. In principle, any measurement of any biological product found in IBD could be defined as a potential “marker” of the disease if it can help in the differential diagnosis or clinical activity of UC or CD. In practice, however, the word “biomarker” is traditionally restricted to those products that have been carefully tested, measured and demonstrated to be actually useful in the clinical setting. Thus, most of the “markers” listed in Table 1 are not true biomarkers, such as ferritin, fibronectin, haemopexin, … sialic acid,.. complement,…interleukin 2,…TL1alpha,…platelet count,…etc. etc., and only some are, such as CRP, … ASCA, ANCA,…calprotectin. Lumping them all together as “biomarkers” is improper and misleading.

  • We agree with the reviewer’s note, and we apologize for the imprudence. In the new version of the manuscript we carefully checked for this mistake.

Page 2:

The sentence “From an immunological standpoint, …. by natural killer T cells” is no longer valid or correct. In fact, the importance of IL-13 to UC pathogenesis has been dismissed by clinical trials showing no clinical benefits by the administration of anti-IL-13 monoclonal antibodies.

The paragraph on the enteric nervous system is disproportional long compared to its known relevance to IBD pathophysiology.

Page 3 and 4:

The whole section “2. Epithelial biology in the gastrointestinal system” is poorly structured and provides scattered and incomplete information. It tries to provide an all-inclusive picture of what goes on in IBD pathogenesis, which should not be part of the scope of the review.

  • To accomplish the reviewer’s suggestion, in the new version of the manuscript we deleted the paragraph 'Epithelial biology in the gastrointestinal system'

The whole sentence “Moreover, epithelial cells… amplify the inflammatory response” is misinforming because it suggests that epithelial cells are the main source of TNF-alpha, which is not the case (macrophages, monocytes and T cells are), and that other pro-inflammatory products are subsequently released; in reality, there are no studies proving which cytokines and cells comes first or which follow, and it is likely that most pro-inflammatory molecules are released concomitantly.

  • See the above comment.

Page 4 to 8:

The sections “3. Antibodies, blood and serum biomarkers” and “4. Stool markers” suffer from the same drawbacks and limitations of the previous section 2. It is also poorly structured and provides scattered and skewed information. Moreover, some of the information is no longer of any value such as the ALCA, ACCA, ASMA and AMCA, because of lack of clinical value and clinical use.

  • According to the reviewer’s suggestion, we better structured the entire paragraph regarding IBD biomarkers.

Page 8 to 13:

The section “5. New biomarkers from mass spectrometry” represents a list of more recent findings in proteomic and lipidomic analyses, the areas of expertise of the authors. Although detailed and more coherent, does not really provide new insights on how these studies can improve current knowledge of IBD pathophysiology.

  • We thank the reviewer for the note. We tried to furnish a more informative section.

ADDITIONAL COMMENTS

Figure 1:

The legend states “CD… is usually located in the lower part of the small bowel and the upper colon”, but this is not shown in the figure, where the terminal ileum and ascending colon are not inflamed.

  • According to the reviewer’s suggestion, we modified Figure 1.

DAMPS are mentioned in the legend but nowhere in the text.

  • We eliminated the acronyms in the legend to the figure.

“injured epithelium cell” should be “injured epithelial cell”.

  • Done

“Goblet cells” should be “goblet cells”.

  • Done

Page 2:

“UC is characterized by a continuous inflammation often limited to the epithelial mucosa”: there is no such a thing as the “epithelial mucosa”, but eventually “mucosal epithelium”; the sentence should actually read “UC is characterized by a continuous inflammation limited to the mucosa and submucosa”.

  • We thank the reviewer for the suggestion. Accordingly, we modified the text

Table 1:

Each listed marker should be referenced as in Table 2.

  • Accordingly, we corrected Table 1.

Line 144:

“accomplished” should be “accompanied” or “associated with”.

  • Done

References:

Many of the references, particularly those that cite findings on cytokines and other immune/inflammatory products in IBD, are from the 1990’s, and many better and more recent comprehensive publications on these topics are available.

  • Accordingly, we inserted more recent publications.

Round 2

Reviewer 1 Report

The authors have successfully addressed most of our previous concerns . However we would like to make sure that the new Figure 2 is appropriately licensed/copyrighted. Particularly, we note that the 'gut associated microbiota' sub-image appears to be a poorly cropped version of another image.

Author Response

We thank the reviewer for the note. Accordingly, we modified the figure 2.

Reviewer 2 Report

GENERAL

The authors have attempted to modify the manuscript by answering to most of the comments and suggestions provided in the first review process. In general, the manuscript is somewhat improved by having provided a more appropriate title, condensed the review of IBD etiopathology, refocused the IBD biomarkers section, redone same tables, and providing a cleaner section on proteomics and lipidomics. Many of the old references have been eliminated and replaced with more recent ones.

On other hand, much work remains to be done, and the authors have not recruited a co-author who is an expert in IBD that could help to better introduce, enumerate and discuss all aspects related to IBD pathophysiology and biomarkers. A series of detailed suggestions and comments are provided below in a very specific way to help the authors to further improve their communication.

SPECIFIC

Line 4

A new author has been added, but no justification has been provided or what he has contributed to.

Line 6-9

Institutes are listed (2, 2) but not the relative affiliation of each author.

Line 13-15

Delete “molecular”, delete “perhaps”, change “correlated to” to “induced by”, replace “bacterial” with “microbial”.

Line 26

Delete “hopeful”.

Line 40-41

The characteristic exacerbations apply to both CD and UC.

Line 46

Delete “perhaps”.

Line 49

There are no pathogenic bacteria in the human intestinal lumen, but eventually pathobionts.

Replace “pass” with “translocate”.

Line 53-57

This whole sentence with the suggested sequence of events is still incorrect. As pointed out in the previous review, the sentence is misinforming because it suggests that epithelial cells are the main source of TNF-alpha, which is not the case (macrophages, monocytes and T cells are), and that other pro-inflammatory products are subsequently released; in reality, there are no studies proving which cytokines and which cells are first involved or which follow, and it is likely that most pro-inflammatory molecules are released concomitantly. The authors should consult with an IBD expert to re-write the whole sentence.

Figure 2.

Only IL6 and calprotectin can be considered biomarkers. The others are inflammatory mediators, not biomarkers.

Line 71

Delete “molecular”.

Line 97-100

Cannot state the “most important” or the “second best”. This has never been established.

“heterogenic”? what does this mean?

Line 104

“unequivocal antigens”? what does this mean?

Line 115-117

Delete the whole sentence on ALCA, ACCA, AMCA.

Line 123

Delete “in adults”.

Table 1

In the Diagnostic column all qualifications are not always correct or reliable, and it should be deleted; also delete the ALCA, ACCA and ASMA rows.

Line 133-134

Replace “flowed into” with “led to”.

Line 142

Replace “following” with “below”.

Line 151

The sentence “useful in establishing correct diagnoses and differentiation of UC and CD patients” is incorrect; none of these studies establish diagnosis or differentiation of CD and UC, they simply attempt to discriminate between the two forms of IBD.

Line 189

“highlining” should be “highlighting”.

Line 198

Delete “successful”.

Line 220

Focusing “on”…

Line 277-285

This sentence is way too long and complex, it must be condensed and simplified.

Line 374

“showing excellent histological specificity and tissue classification” what does this mean?

Author Response

We thank the reviewer for the punctual revision he made. We corrected all the raised indications.
Moreover, we have now asked for help from a clinician (Dr. Chieppa), partner of the project we acknowledge at the end of the manuscript. We hope that its contribution has been helpful for the improvement of the manuscript.
